# Extracellular Vesicles and Antiphospholipid Syndrome: State-of-the-Art and Future Challenges

**DOI:** 10.3390/ijms22094689

**Published:** 2021-04-28

**Authors:** Ula Štok, Saša Čučnik, Snežna Sodin-Šemrl, Polona Žigon

**Affiliations:** 1Department of Rheumatology, University Medical Centre Ljubljana, SI-1000 Ljubljana, Slovenia; sasa.cucnik@kclj.si (S.Č.); snezna.sodin@kclj.si (S.S.-Š.); polona.zigon@guest.arnes.si (P.Ž.); 2Faculty of Pharmacy, University of Ljubljana, SI-1000 Ljubljana, Slovenia; 3Faculty of Mathematics, Natural Sciences and Information Technologies, University of Primorska, SI-6000 Koper, Slovenia

**Keywords:** extracellular vesicles, antiphospholipid syndrome, antiphospholipid antibodies, thrombosis, preeclampsia, endothelial cells, monocytes, platelets, placental explants, trophoblasts

## Abstract

Antiphospholipid syndrome (APS) is a systemic autoimmune disorder characterized by thromboembolism, obstetric complications, and the presence of antiphospholipid antibodies (aPL). Extracellular vesicles (EVs) play a key role in intercellular communication and connectivity and are known to be involved in endothelial and vascular pathologies. Despite well-characterized in vitro and in vivo models of APS pathology, the field of EVs remains largely unexplored. This review recapitulates recent findings on the role of EVs in APS, focusing on their contribution to endothelial dysfunction. Several studies have found that APS patients with a history of thrombotic events have increased levels of EVs, particularly of endothelial origin. In obstetric APS, research on plasma levels of EVs is limited, but it appears that levels of EVs are increased. In general, there is evidence that EVs activate endothelial cells, exhibit proinflammatory and procoagulant effects, interact directly with cell receptors, and transfer biological material. Future studies on EVs in APS may provide new insights into APS pathology and reveal their potential as biomarkers to identify patients at increased risk.

## 1. Introduction

Antiphospholipid syndrome (APS) is a systemic autoimmune disorder characterized by thrombosis and/or obstetric complications with persistent presence of antiphospholipid antibodies (aPL) [1]. aPL are a heterogeneous group of autoantibodies, of which anti-cardiolipin (anti-aCL), anti-β2 glycoprotein I (anti-β2GPI), and lupus anticoagulant (LA) are included in the laboratory criteria for the diagnosis of APS [2]. It has been found that in addition to criterion aPL, other non-criterion aPL, such as antibodies against phosphatidylserine/prothrombin complex, also play an important role in APS [3,4]. Extracellular vesicles (EVs) are submicron particles that are continuously released from nearly all cell types under physiological conditions, and circulate in the plasma of healthy individuals at concentrations of approximately 10^10^ EVs/mL [5]. The classification and also the nomenclature of EVs are complicated and could be confusing due to overlapping definitions. The most common classification of EVs used in the literature is the division of different EVs into subtypes, such as endosomal-derived exosomes (nanovesicles), membrane-derived microvesicles (microparticles or ectosomes), and apoptotic bodies. This classification is based on the assignment of a specific EV to a particular biogenesis pathway, which remains very difficult to assess [6]. Unless biogenesis is investigated directly, EVs are classified according to their (a) physical characteristics such as size: “small EVs” (sEVs; size <100 nm or <200 nm) and “medium/large” (m/lEVs; size >200 nm) and density: low, medium, or high, with defined range; (b) biochemical composition (surface expression or by the presence of a specific molecule within EVs); or (c) description of a specific condition or cell of origin [6].

The key biological function of EVs is cell-to-cell communication and the transfer of biological materials that act closely, but also, and more importantly, remotely. Cargo within EVs is protected from degradation in the bloodstream and can be successfully transferred to specific cells of interest, affecting several biological functions of these cells. EVs can transfer a wide variety of molecules including heat shock proteins (HSP-90, HSP-70), cytokines, such as interleukins (ILs), and tumor necrosis factor-alpha (TNFα), enzymes, peptides, growth factors, RNA, including micro RNA (miR), and DNA [7]. Given the fact that EVs migrate through the bloodstream, they can have pleiotropic effects that can affect many tissues in the body. In response to stimuli, such as cell activation due to inflammation and/or apoptosis, or pregnancy, increased amounts of EVs are released. An increase in circulating EVs, especially endothelial EVs, is considered one of the hallmarks of vascular dysfunction and cardiovascular disease. Increased EVs are found in patients with arterial and venous thrombosis, pulmonary embolism, hypertension, diabetes, acute coronary syndrome, and some other cardiovascular disorders [8,9]. The physiological release of EVs is usually increased during pregnancy, and EVs have been shown to carry or express increased levels of proinflammatory and procoagulant molecules on their surface, which contribute to an enhanced inflammation and hypercoagulation state significant for pregnancy [10]. However, there are several studies suggesting that elevated concentrations and altered profiles of EVs play a role in pregnancy disorders, such as preeclampsia (PE), gestational diabetes mellitus (GDM), recurrent fetal loss, and preterm birth [11,12,13].

This review provides an up-to-date overview of the role of EVs in vascular and obstetric pathologies, with a focus on APS. In contrast to previous reviews [14,15,16], this review separately describes and examines the characteristics of EVs isolated from plasma of both thrombotic and obstetric APS patients, as well as includes studies examining the in vitro effects of aPL on EV release.

## 2. The Molecular Mechanisms of EVs Contributing to Vascular Disorders

Endothelial cells, platelets, and monocytes are key players in the maintenance of vascular hemostasis, at least in part, by the release of EVs. EVs are continuously released from cells into the extracellular environment; however, during pathology, their levels and composition are significantly changed. Hemostasis is a very strictly regulated process that maintains normal function of vasculature despite the presence of triggers, such as injury and/or infection. In pathology, EVs alter the hemostatic balance by increasing inflammation, coagulation, and endothelial dysfunction and contribute to the development of various pathologies, including deep vein thrombosis or pulmonary embolism [9] and cardiovascular disorders [8].

### 2.1. Platelet EVs Change the Adhesion Profile of Endothelial Cells and Monocytes, Activate Other Platelets, and Influence Cytokine Production

EVs extruded from activated platelets have various effects on endothelial cells, monocytes, and other platelets (Figure 1A). Specifically, increased levels of intracellular adhesion molecule-1 (ICAM-1) [17,18] were observed on endothelial cells after stimulation with platelet EVs, an effect later ascribed to miR-320b transfer [17]. Stimulation with platelet EVs also increased the expression of lymphocyte function-associated antigen-1 LFA-1 (CD11a/CD18) and macrophage antigen-1 Mac-1 (CD11b/CD18) on monocytes. This is a process dependent on the arachidonic acid transfer from platelet EVs and consequent activation of protein kinase C [19]. Platelet EVs therefore significantly modulate monocyte–endothelial interactions, as well as leukocytes interactions [20,21], and contribute significantly to increased adhesion and aggregation of platelets [22,23] and leukocytes to blood vessel walls during pathology, including APS. In addition, platelet EVs were shown to influence cytokine production (IL-1β, IL-6, IL-8) [18] and the transfer of miRNA (miRs 142-3p and 223). MiR-142-3p has been found to be required for immunomodulatory and regulatory functions of endothelial progenitor cells [24] and has been described to modulate various inflammatory responses [25]. Platelet-derived EVs promote endothelial cell proliferation via miR-142-3p [26], while miR-223 [27] promotes endothelial cell apoptosis. In addition, thrombin-activated platelet-derived EVs regulate endothelial cell activation through ICAM-1 expression via miR-223 during the thrombosis-inflammation response [28]. These miRNAs are thereby affecting endothelial cell activation, proliferation, and apoptosis.

Importantly, there is also evidence that platelet EVs have certain anticoagulant effects [29,30], for instance, platelet-derived EVs inhibit atherothrombotic processes by reducing CD36-dependent lipid loading of macrophages and by suppressing platelet thrombosis. Further research is needed to determine which key stimuli are responsible for determining whether platelet EVs have pro- or anticoagulant effects.

### 2.2. Endothelial EVs Carry a Proadhesive and Procoagulant Profile

Endothelial dysfunction, an alteration in endothelial phenotype, is a hallmark of many vascular pathologies, including APS, and endothelial EVs have been suggested to play an important role (Figure 1B). Various proinflammatory stimuli such as TNF-α, lipopolysaccharide, C-reactive protein, reactive oxygen species, and coagulation stimuli such as thrombin and plasminogen activator inhibitor-1 (PAI-1) increase the level of endothelial EVs. These EVs carry adhesion molecules (ICAM-1, vascular cell adhesion protein 1 (VCAM-1), E-selectin, VE-cadherin, α-integrin), coagulation factors (von Willebrand factor (vWF), tissue factor (TF), PAI-1), and growth factors (endoglin, CD146, vascular endothelial growth factor (VEGF) receptor) [31,32,33], which induce interactions with cells and provide a procoagulant surface for the binding of coagulation factors. Endothelial EVs can also induce the expression of TF on monocytes [34]. In addition, endothelial EVs enhance endothelial dysfunction by attenuating the production of nitric oxide from endothelial cells [35]. Importantly, endothelial EVs may also have anticoagulant and anti-inflammatory potential [8]; however, these vesicles are generally believed to impair vascular function [36].

### 2.3. Monocyte EVs Modulate Adhesion and Coagulation Profile of Endothelial Cells

Upon their activation, monocytes release increased levels EVs, which contribute to disruption of the hemostatic balance, which is also a key feature in the pathogenesis of APS (Figure 1C). Monocyte EVs significantly alter the adhesion profile of endothelial cells. After initial interaction via LFA-1-ICAM-1 [37], EVs are internalized into endothelial cells and were shown to regulate the expression of ICAM-1, VCAM-1, and E-selectin via extracellular signal-regulated protein kinase (ERK1/2) and nuclear factor-κB (NF-κB) signaling pathways [38]. In addition, monocyte EVs also appear to induce de novo synthesis of ICAM-1, chemokine C-C motif ligand 2 (CCL2), and IL-1β in endothelial cells via activation of toll-like receptor 4 (TLR4)/Myeloid differentiation primary response gene 88 (MyD88)/NF-κB signaling pathway [39]. The modified endothelial cell adhesion profile makes these cells more prone to interactions with leukocytes and platelets, increasing the prothrombotic state of the vasculature. Monocyte EVs also trigger proinflammatory signaling pathways by transferring immunomodulatory miRs to recipient cells. Levels of miR-125a-5p, miR-146a, miR-146b, and miR-155 were significantly increased in EVs derived from IFNα- and lipopolysaccharide-stimulated monocytes compared to EVs released from unstimulated monocytes, while the levels of miR-222 levels were decreased [40]. miR-222 acts protectively by controlling endothelial inflammatory activation and proliferation [41]. A decrease in the endothelial miR-221/222 cluster has been reported as a contributing factor in various vascular disorders including coronary artery disease, heart failure, hypertension, obesity, and atherosclerosis [41,42]. A primary cellular initiator of coagulation, TF, was found to be expressed on both monocytes and monocyte EVs [37]. During vascular injury, TF forms a complex with factor VIIa, which activates the coagulation protease cascade, eventually leading to fibrin deposition and platelet activation [43]. Monocyte EVs have also been found to decrease the levels of the anticoagulant TFPI and thrombomodulin on endothelial cells [37].

## 3. The Molecular Mechanisms of EVs Contributing to Pregnancy Disorders

Extracellular vesicles are involved in fetal–maternal communications and as such are an important part of the physiological processes that occur during normal pregnancy, including embryo implantation, immune regulation, spiral artery remodeling, metabolic adaptations, and parturition [13]. EVs are continuously released from cells to the extracellular environment, but their amount and composition are significantly altered during pathology. In addition to EVs normally present in human plasma, trophoblast EVs are present only during pregnancy. Three subtypes of trophoblasts can be found in the human placenta: villous cytotrophoblasts, extravillous trophoblasts (EVT), and syncytiotrophoblasts (STB). The STB is a multinucleated single cell that covers the maternal-facing part of the placenta, plays a key role in the exchange of materials between mother and fetus, and is the main site of EV production and release into the maternal bloodstream [16]. In addition to conventional EVs, trophoblasts also release trophoblastic debris, which contain an average of 60 nuclei [44] and range in size from 20–500 μm [44,45]. In pathology, EVs alter the normal pregnancy balance by increasing inflammation, coagulation, and endothelial dysfunction, contributing to the development of pregnancy disorders such as PE, GDM, recurrent fetal loss, and preterm birth [11,12,13].

### Trophoblast EVs and EVs from Plasma of Patients with Obstetric Complications Enhance Inflammation, Endothelial Dysfunction, and Hypercoagulation

To investigate the molecular mechanisms of EVs in correlation with obstetric complications, EVs obtained from the plasma of obstetric patients or from cultured trophoblasts were used. A proinflammatory profile is normally present during pregnancy; however, dysregulation of inflammation is further increased in pregnancy-related disorders in which EVs were found to play a role. EVs extruded from preeclamptic placental explants stimulated monocytes to release proinflammatory factors (IL-1β, IL-6, IL17, CSF3, CCL3, CCL4, CCL5, TNFα, CXCL1). In general, compared to normal placental EVs, preeclamptic EVs induced an exaggerated proinflammatory response by suppressing monocytes chemotactic activity and motility [46,47]. Furthermore, EVs isolated from plasma of GDM patients increased the release of proinflammatory cytokines (GM-CSF, IL6, IL8) from endothelial cells [48] similar to EVs from placental explants cultured under high glucose [49]. EVs derived from EVT cultured under hypoxic conditions decreased endothelial cells migration and increased TNFα release from these cells. Cargo analysis from these EVs revealed miRNAs associated with cell migration and cytokine production. Notably, three miRNAs (miR-1269b, miR-525-5p, and miR-526b-5p) were identified only in EVs derived from EVT cultured under hypoxic conditions and in EVs derived from plasma of PE patients and patients with spontaneous preterm birth [50]. Moreover, miR-141 was enriched in EVs from preeclamptic placentae compared to EVs from normal placentae. Weimar et al. showed that overexpression of miR-141 in trophoblastic cells is also reflected in their EVs, which inhibit T cell proliferation [51]. MiR-548-5p is an anti-inflammatory factor that regulates macrophages’ activation and proliferation. Levels of miR-548-5p are decreased in both EVs and peripheral mononuclear cells isolated from the serum of PE patients, which may have an impact on the enhanced inflammatory state [52] (Figure 2A).

Impaired endothelial function and vasoconstriction significantly increases the risk of vascular complications in pregnancy, especially in PE [53,54]. Fms-like tyrosine kinase-1 (Flt-1), a known anti-angiogenic factor, is present on the surface of STB EVs and has been suggested to be involved in endothelial injury induced by STB EVs in PE patients [55,56]. Soluble Flt-1 and soluble endoglin (sEng) were increased in EVs isolated from plasma of PE patients compared to normal controls. In addition, Chang et al. showed that EVs mediate efficient transfer of sFlt-1 and sEng to endothelial cells, which attenuates proliferation, migration, and tubule formation of these cells in vitro [57]. Placenta-specific C19MC miRNAs derived from STB EVs have been shown to be transferred to endoplasmic reticulum and mitochondria of endothelial cells, downregulating specific target genes including Flt-1 [58]. The expression of adhesion molecules, such as ICAM-1, VCAM-1, E-selectin, F-selectin, and vitronectin, on EVs extruded from placental explants could affect the adhesion of these EVs to various target cells and promote their interactions [10]. The decreased eNOS level is associated with decreased availability of NO in PE. EVs isolated from plasma of PE patients as well as EVs from perfusate of placenta explant have lower eNOS levels [59]. Treatment of endothelial cells with EVs from serum of PE patients reduced the expression of eNOS and consequently NO production in endothelial cells. The levels of miR-155 were found to be increased in EVs from PE patients compared to controls. In vitro studies showed that miR-155 can be transferred from EVs to endothelial cells and is able to suppress eNOS expression [60] (Figure 2B).

A hypercoagulable state often develops in normal pregnancy but is much more pronounced in PE, leading to microvascular thrombosis and organ ischemia [61]. It has been found that procoagulant molecules such as TF and PS are present on the surface of STB EVs and can induce a systemic hypercoagulable state found in PE patients [62,63]. PAI-2, which is involved in the regulation of fibrinolysis, was detected on STB EVs from placental lobe dual perfusions [64] (Figure 2B).

## 4. Extracellular Vesicles in Antiphospholipid Syndrome

### 4.1. Literature Search Strategy and Results

A literature search was performed in the MEDILINE (PubMed) database using the following key words: (Antiphospholipid syndrome OR antiphospholipid antibody*) AND (extracellular vesicle* OR exosome* OR microparticle* OR microvesicle* OR ectosome* OR trophoblast debris). In this literature review, the most common terms to define EVs were used. However, due to the heterogeneity of nomenclature, we allow the possibility that some studies might have been overlooked.

The search was limited to articles that were available in full text, written in English, included APS population, and whose topic was related to the investigation of EVs. Commentaries on articles, editorials, case reports, review articles, reports investigating other patient groups (mainly SLE, SSC, or RA patients), or reports focusing on another diseases or topic (not EVs) were excluded. The search included articles from 1992 to 2021.

The literature search, performed on 3 February 2021, yielded 101 records. Twenty-eight articles that met the inclusion criteria were ultimately included in the review; 16 studies focused on thrombotic APS, 11 on obstetric APS, and one study included both. Out of 73 excluded articles, 35 were review articles (15 involved EVs related to aPL or APS, 14 were not fully related to the topic, and six described pathologies other than APS). The remaining 33 articles were original articles (14 were not fully related to the topic, eight investigated SLE patients, nine described pathologies other than APS, and two were not written in English); one was a case report, three were commentaries, and one was an editorial.

### 4.2. Thrombotic APS

EVs are associated with many vascular pathologies, including arterial thrombosis and deep venous thrombosis associated with pulmonary embolism, both of which are part of the clinical characterization of thrombotic APS [14,15]. The role of EVs in the pathology of APS has been studied in vivo by isolating EVs from patient plasma and in vitro on EVs isolated from aPL-stimulated cell cultures. Most commonly, the characterization of EVs from plasma of APS patients is based on the determination of their cell origin and prothrombotic profile (e.g., by the presence of TF and PS). In vitro systems are used to further characterize EVs released upon aPL stimulation of endothelial cells and platelets and to determine possible downstream effects these EVs have on target cells. Because of heterogeneity in the nomenclature used to define different EVs, different studies have used different terms, small EVs (alternatively called nanovesicles or exosomes) and medium to large EVs (alternatively called microvesicles or microparticles).

To date and to our knowledge, 12 studies investigated EVs in plasma from thrombotic APS patients (Table 1), four studies investigated in vitro effects of EVs derived from aPL-stimulated endothelial cells or platelets (Table 2), and one study was enrolled in both clinical and translational studies.

#### 4.2.1. Thrombotic APS In Vivo (Clinical) Studies

Most studies investigated medium/large EVs of endothelial origin, as endothelial dysfunction is one of the most important features of APS. Elevated levels of endothelial (CD51+) EVs have been shown in aPL+ patients compared to HBDs [31]. Increased levels of endothelial EVs (CD31+, CD51+, CD105+, CD144+) in the plasma of APS patients compared to HBDs were later confirmed in several other studies [65,66,67,68,69,70], with one exception where the increase was not observed [71] (Table 1).

Moreover, a significant increase in endothelial EVs was observed in aPL+ thrombotic patients compared to asymptomatic aPL+ patients [31], suggesting that thrombosis rather than aPL affects the release of EVs, which was later also confirmed in two other studies [69,70]. In contrast, Jy et al. found no difference in the levels of endothelial EVs (CD31+/CD42-) between aPL+ thrombotic patients and asymptomatic aPL+ patients [66], suggesting that the release of EVs is more related to the autoimmune process involving the presence of aPL than to thrombosis itself. Latter was supported also by Dignat-George et al., where they showed an increase in endothelial (CD51+) EVs between APS patients and aPL– patients with thrombosis [65] (Table 1).

Studies that examined platelet or monocyte EVs are inconsistent; while some studies found increased levels of platelet EVs in APS patients compared to HBDs [66,68,69,70,72], others did not [67,71,73], and some reported increased levels of monocyte EVs [67,72], while others did not [68,71] (Table 1). In addition, it is known that aPL induce TF expression on endothelial cells and monocytes, but it is not clear whether this is accompanied by the release of TF + EVs [67,68,74] or not [69,71,75] (Table 1).

**Table 1 ijms-22-04689-t001:** Isolation, quantification and characterization of EVs in plasma of thrombotic APS patients.

Reference	Patient Group	Control Group	Isolation Method	Characterization Method	Type of EVs	Main Findings
Combes et al., 1999 [31]	5 APS, 8 APS + SLE	17 asympt. aPL+ (6 SLE or other autoimmune, 4 infections, 5 malignancies, 2 undefined) 30 HBDs	Sodium citrate 2 × 1500× *g* (15 min), 13,000× *g* (1 min). Temperature not specified.	FC: Positive for annexin V, CD51. < 1.5 µm (using latex beads). Renumeration beads not specified.	endothelial (CD51+)	↑ levels of endothelial EVs in aPL + pts. vs. HBDs. ↑ levels of endothelial EVs in aPL + pts. vs. asympt. aPL+.
Joseph et al., 2001 [73]	20 APS 14 APS + SLE	16 SLE 20 HBDs	Sodium citrate 2 × 1500× *g* (15 min), 13,000× *g* (1 min). Temperature not specified.	FC: Positive for GPIIb-IIIa. < 0.8 µm (beads not specified). Renumeration beads not specified.	platelet (GPIIb-IIIa+)	No difference in levels of platelet EVs between APS pts., SLE pts., and HBDs.
Nagahama et al., 2003 [72]	24 APS 13 APS + SLE	25 SLE 30 HBDs	Sodium citrate 200× *g* (10 min, RT), 1000× *g* (15 min, RT).	FC: Positive for annexin V, CD14, CD42a.	platelet (CD42a+) monocyte (annexin V+/CD14+)	↑ levels of monocyte EVs in APS pts. vs. APS + SLE pts. and vs. HBDs. ↑levels of P-selectin+ platelets and platelet EVs in APS pts. vs. SLE pts. and HBDs.
Dignat-George et al., 2004 [65]	23 APS 14 APS + SLE	28 SLE aPL+ no thrombosis 23 SLE aPL− no thrombosis 25 aPL− with thrombosis 25 HBDs	Sodium citrate 2 × 1500× *g* (15 min), 13,000× *g* (2 min). Temperature not specified.	FC: Positive for CD51. <0.8 µm (using latex beads). Renumeration beads (FlowCount).	endothelial (CD51+)	↑ levels of endothelial EVs in APS pts. vs. HBDs and vs. aPL− thrombosis. No difference between primary or secondary APS. ↑ levels of endothelial EVs SLE aPL + pts. vs. HBDs. No difference between SLE aPL− pts. and aPL− thrombosis pts. vs. HBDs.
Jy et al., 2007 [66]	60 APS	28 asympt. aPL+ 39 HBDs	Sodium citrate 160× *g* (10 min), 1500× *g* (6 min). Temperature not specified.	FC: Positive for CD31, CD42. < 1.5 µm (beads not specified). Renumeration beads (Standard beads).	endothelial (CD31+/CD42-) platelet (CD31+/CD42+)	↑ levels of platelet and endothelial EVs in APS pts. vs. HBDs. ↑ levels of endothelial EVs in asympt. aPL+ vs. HBDs. ↑ levels of platelet EVs in APS pts. vs. asympt. aPL+. No difference in levels of endothelial EVs in APS pts. vs. asympt. aPL+. No difference in levels of platelet EVs in asympt. aPL+ vs. HBDs.
Flores-Nascimento et al., 2009 [71]	11 APS 9 DVT pts. at diagnosis 10 DVT pts. after 1–3 years of warfarin withdrawal 7 FVL pts.	37 HBDs	Sodium citrate 3000× *g* (20 min), 13,000× *g* (30 min). Temperature not specified.	FC: Positive for annexin V, CD14, CD31, CD45, CD61, CD142, CD235.	total (annexin V+) platelet (CD61+) erythrocyte (CD235+) monocyte (CD14+) endothelial (CD31+) leukocyte (CD45+) TF (CD142+)	No difference in total EVs in DVT pts. at diagnosis, FVL pts., APS pts. vs. HBDs. ↑ levels of total EVs in DVT 1-3 years vs. HBDs. No difference in platelet, erythrocyte, monocyte, endothelial, and leukocyte EVs in all pts. groups vs. HBDs.
Vikerfors et al., 2012 [67]	40 APS 12 APS + SLE	52 HBDs	Blood collection and centrifugation not described.	FC: Negative for phalloidin, positive for lacadherin, CD14, CD42a, CD142, CD144. <1 µm (using MegaMix beads). Renumeration beads not specified.	total (lacadherin+) endothelial (CD144+) platelet (CD42a+) monocyte (CD14+) endothelial TF (CD144+/CD142+)	↑ levels of total EVs in APS pts. vs. HBDs. ↑ levels of endothelial, endothelial TF+, and monocyte EVs in APS pts. vs HBDs. No difference in levels of platelet EVs in APS pts. vs. HBDs.
Willemze et al., 2014 [74]	11 APS 19 APS + SLE	72 asympt. aPL+ no HBDs	Sodium citrate 1500× *g* (10 min, 4 °C), 2000× *g* (5 min, 4 °C), 20,000× *g* (30 min, 4 °C).	TF activity assay	TF + EVs	↑ EV-TF activity in APS pts. vs. asympt. aPL+. No difference in EV-TF activity in the presence or absence of underlying SLE. No difference between different APS clinical complications. No correlation between EV-TF activity and aPL subtype.
Breen et al., 2015 [69]	37 APS	18 asymptomatic aPL+, 18 HBDs	Sodium citrate 2 × 2000× *g* (15 min, 4 °C), for procoagulant activity additional 12,000× *g* (2 min, 4 °C).	FC: Positive for CD51, CD41, CD61, CD105. Renumeration beads (flow count fluoroshperes)	endothelial (CD51+ or CD105+) platelet (CD41+ or CD61+)	↑ levels of endothelial and platelet EVs in APS pts. vs. HBDs. No difference in levels of endothelial and platelet EVs in asymptomatic aPL+ vs. HBDs. No difference in the EVs procoagulant activity between all groups and HBDs.
Chaturvedi et al., 2015 [68]	47 aPL+ patients (38 APS, 2 APS + SLE, 6 asympt. aPL+, 1 aPL+ migraine with aura)	144 HBDs	Sodium citrate 2 × 1500× *g* (15 min), 13,000× *g* (2 min). Temperature not specified.	FC: Positive for annexin V or CD14, CD41, CD105, CD142, CD144. < 1µm (using latex beads). Renumeration beads not specified.	total (annexin V+) endothelial (CD105+ or CD144+) platelet (CD41+) monocyte (CD14+) TF (CD142+)	↑ levels of total EVs in aPL+ vs. HBDs. ↑ levels of endothelial, platelet, and TF + EVs in aPL+ vs. HBDs. No difference in levels of monocyte EVs in aPL+ vs. HBDs.
Niccolai et al., 2015 [70]	16 APS	16 asympt. aPL+ 16 HBDs	Sodium citrate Serial centrifugation: 1500× *g* (15 min), 3000× *g* (3 min). Temperature not specified.	FC: Positive for VPD450 or CD31, CD41a, CD45. < 0.9 µm (using Megamix beads). Renumeration beads (Trucount).	total (VPD450+/7AAD-) endothelial (CD31+) platelet (CD41a+) leukocyte (CD45+)	↑ levels of total, endothelial, platelet, and leukocyte EVs between APS pts. and HBDs, between APS pts. and asympt. aPL+ and between asympt. aPL + pts. and HBDs. ↑ levels of total EVs in APS triple positivity vs. single positivity. ↑ levels of endothelial EVs in asympt. aPL+ triple positivity vs. single positivity.
Hell et al., 2018 [75]	64 APS 18 APS + SLE 12 APS+LLD	30 HBDs	Sodium citrate 2500× *g* (15 min, 15 °C)	TF activity assay	TF + EVs	No difference in EV-TF activity in APS pts. vs. HBDs. No difference in EV-TF activity in single, double, or triple aPL + pts. No correlation between different aPL and EV-TF activity. No difference in EV-TF activity in aPL + pts. with arterial thrombosis vs. venous thrombosis vs. combination of both. No difference in EV-TF activity and the number of thromboses.
Štok et al., 2020 [76]	14 APS	5 aPL− with thrombosis 7 HBD	Sodium citrate 820× *g* (10 min, RT), 2500× *g* (10 min, RT), 10,000× *g*, (45 min, RT), 100,000× *g* (2 h 15 min, 4 °C ).	NTA	< 200 nm. multiplex flow cytometry for 38 markers (detection via tetraspanins)	Small EVs were investigated. ↑ levels of sEVs in plasma of APS pts. vs. HBD. Platelet (CD41b+, CD42a+), lymphocyte (CD8+), leukocyte (CD45+), and endothelial (CD31+) sEVs were detected. ↑ levels of P-selectin on sEVs from APS pts. vs. HBDs. ↑levels of CD133/1 on sEVs from APS pts. vs. aPL− pts. with thrombosis.

aPL: antiphospholipid antibodies; APS: antiphospholipid syndrome; asympt.: asymptomatic; DVT: deep vein thrombosis; EVs: extracellular vesicles; FC: flow cytometry; FVL: factor V Leiden; HBDs: healthy blood donors; LLD: lupus like disease; pts.: patients; NTA: nanoparticle tracking analysis; sEVs: small extracellular vesicles; SLE: systemic lupus erythematosus; TF: tissue factor; ↑: elevated levels.

The only study that investigated the presence of sEVs, that is, EVs of smaller size (<200 nm), in the plasma of APS patients found that their levels were significantly elevated compared to HBDs [76] (Table 1). Increased levels of sEVs were observed in both aPL+ APS patients and aPL− patients with a history of thrombosis, suggesting that thrombosis rather than aPL plays a role in the release of platelet EVs. sEVs from APS patients were enriched in surface expression of P-selectin, suggesting endothelial and platelet activation in APS. APS patients also showed increased CD133/1 expression compared to aPL− patients with thrombosis, indicating endothelial damage in APS. There was no difference in the levels of platelet (CD41b+, CD42a+), lymphocyte (CD8+), leukocyte (CD45+), and endothelial (CD31+) sEVs between APS patients and HBDs (Table 1).

The heterogeneity of published reports is at least partly due to a lack of standard in-sample processing, detection reagents, and instrument settings. Some reports include a rather small group of patients and controls and mostly did not define laboratory measurements that have been shown to affect the concentration of EVs. Pre- and post-analytical variability is known to significantly affect EVs’ concentration and their physical characteristics [6].

#### 4.2.2. Thrombotic APS In Vitro (Translational) Studies

Both monoclonal and polyclonal aPL have been shown to activate endothelial cells, monocytes, and platelets contributing to a thrombotic phenotype [77,78]. Furthermore, the release of EVs after aPL stimulation has been studied in endothelial cells and platelets [65,79,80,81] (Table 2), but to our knowledge not in monocytes.

**Table 2 ijms-22-04689-t002:** Isolation, quantification and characterization of EVs derived from endothelial cells and platelets after stimulation with aPL.

Reference	Cell Type	Stimulation	Isolation Method	Characterization Method	Main Findings
Ford et al., 1998 [81]	Platelets	Serum from aPL + pts. and HBDs.	No isolation.	FC: Platelet EVs (CD61+).	No difference in levels of CD61+ vesicles after stimulation of platelets with aPL + pts. serum vs. HBDs.
Dignat-George et al., 2004 [65]	HUVEC	Plasma from APS pts. and HBDs.	None for flow cytometry. 14,000× *g* (2 h 30 min) for procoagulant activity.	FC: Total EVs (annexin V+). <0.8 µm (using latex beads). Renumeration beads (Flowcount).	↑ levels and procoagulant activity of EVs after stimulation with APS pts. plasma vs. HBDs.
Wu et al., 2015 [82]	HUVEC	Anti-β2GPI from APS pts. and rabbits immunized with β2GPI. Control IgG.	2500× *g* (15 min), 13,000× *g* (2 min), 100,000× *g* (90 min).	qPCR, immunoblotting.	↑ activation of endothelial cells after stimulation with endothelial EVs released in response to aPL possibly through presentation of ssRNA (miRNA) to TLR7. ↑ active IL-1β in endothelial EVs.
Pericleous et al., 2012 [79]	HUVEC	Polyclonal IgG from APS pts. and HBDs.	3000× *g* (5 min), 12,000× *g* (60 min).	FC: Total EVs (annexin V+) Specific endothelial EVs: CD62E+ (E-selectin), CD106+ (VCAM-1), CD54+ (ICAM-1), CD142+ (TF), CD105+ (endoglin), CD144+ (VE-cadherin). <1 µm (using latex beads).	↑ levels of total endothelial and E-selectin+ EVs after APS IgG stimulation vs. HBD IgG. No difference in levels of ICAM-1+, endoglin+, and VE-cadherin+ EVs after APS IgG stimulation vs. HBD IgG. VCAM-1+ and TF + EVs could not be detected.
Betapudi et al., 2013 [80]	HUVEC	Anti-β2GPI purified from APS pts. and rabbits immunized with β2GPI. Control IgG.	1500× *g*, (30 min), 13,000× *g* (2 min).	FC: <1 µm (using latex beads ). Endothelial EVs (CD144+).	↑ levels of endothelial EVs after stimulation with aPL compared to control IgG. Anti-β2GPI antibodies stimulate EVs release via a nonmuscle myosin II motor protein-dependent pathway.

aPL: antiphospholipid antibodies; APS: antiphospholipid syndrome; β2GPI: β2-glycoprotein I; EVs: extracellular vesicles; FC: flow cytometry; HBDs: healthy blood donors; HUVEC: human umbilical vein endothelial cells; ICAM-1: intercellular adhesion molecule 1; IgG: immunoglobulin G; IL: interleukin; miRNA: micro RNA; NA: not analyzed; pts: patients; ssRNA; single stranded RNA; TF: tissue factor; TLR: toll-like receptor; VCAM-1: vascular cell adhesion molecule 1; ↑: elevated levels.

Studies investigating medium/large EVs showed that stimulation of endothelial cells with aPL [79,80] or plasma from APS patients [65] resulted in an increased release of endothelial EVs compared to control stimulation (Table 2). A study by Dignat-George et al. showed a significant four-fold increase in endothelial EVs with procoagulant activity after stimulation of endothelial cells with plasma from APS patients [65], whereas only a moderate increase was observed when endothelial cells were stimulated with HBDs plasma. Another study also observed a two-fold increase in levels of endothelial EVs released by endothelial cells stimulated with anti-β2GPI [80]. Stimulation of endothelial cells with aPL caused the release of EVs enriched in active IL-1β, which had a distinct miRNA profile and caused endothelial cell activation [82]. However, endothelial cell activation appeared to be mediated by the TLR/myd88-IRAK4 signaling pathway, rather than by the IL-1β receptor. The authors concluded that at least one mechanism by which EVs induce cellular activation is through the presentation of ssRNA, most likely miRNA, to TLR7 [82] (Table 2).

Investigating the surface protein profile of EVs released from aPL-stimulated endothelial cells revealed that a greater proportion of these EVs carried surface E-selectin, whereas the levels of ICAM-1+, endoglin+, and VE-cadherin+ EVs did not differ from control stimulation. In addition, VCAM-1+ and TF+ endothelial EVs could not be detected [79] (Table 2). The only study investigating platelet stimulation with serum from APS patients did not show increased release of platelet (CD61+) EVs compared to stimulation with serum from HBDs [81] (Table 2).

EVs are released by endothelial cells in response to stimulation with aPL and appear to have a distinct protein and RNA profiles. Current research is limited and heterogeneous in stimulation and EVs isolation conditions. Further research is needed to support the findings on EVs characteristics and to determine the mechanisms of their release and influence on other target cells.

### 4.3. Obstetric APS

EVs are normally released during pregnancy and carry or express increased levels of proinflammatory and procoagulant molecules on their surface, which contribute to enhanced inflammation and hypercoagulation during pregnancy [10]. Elevated levels of EVs and their distinct profile have been suggested to play a role in pregnancy disorders including PE and recurrent fetal loss, both of which are part of the clinical manifestations of obstetric APS [10,16]. The role of EVs in the pathology of obstetric APS has been investigated in vivo by isolating EVs from patient plasma and in vitro on EVs isolated from aPL-stimulated cell cultures. The characterization of EVs from plasma of APS patients is based on determining their cell of origin, their surface profile, or by studying their cargo. In vitro systems are used to study the release of EVs after aPL stimulation of placental explants and to determine possible downstream effects on target cells. Due to heterogeneity in the nomenclature used to define different EVs, studies have used diverse terms; small EVs (alternatively termed nanovesicles or exosomes), medium to large EVs (alternatively termed microvesicles or microparticles), and trophoblastic debris (alternatively termed macrovesicles).

To date and to our knowledge, four studies investigated EVs in plasma from obstetric APS patients (Table 3), seven translational studies investigated the in vitro effects of EVs on endothelial cells either derived from placental explants or isolated from plasma of patients with APS obstetric complications (Table 4), and one study was enrolled in both clinical and translational studies.

#### 4.3.1. Obstetric APS In Vivo (Clinical) Studies

Studies investigating EVs in APS patients with a history of obstetric complications enrolled either pregnant [83,84,85] or non-pregnant APS patients [69,86]. In one study of non-pregnant obstetric APS patients, increased levels of total EVs were observed compared to healthy non-pregnant women [86]; however, in another study, Breen et al. did not detect increases in specific EVs populations, endothelial (CD51+ or CD105+) and platelet (CD41+ or CD61+), between the two groups [69]. Levels of total EVs were comparable between both pregnant and non-pregnant aPL+ and aPL− patients, suggesting that levels of EVs are not associated with aPL [83,86]. Furthermore, there was no difference in endothelial (CD144+), platelet (CD41+), and leukocyte (CD45+) EVs between aPL+ and aPL− pregnant patients with a history of pregnancy loss [83] (Table 3). These results might be partially due to a very small sample size of aPL+ patients.

**Table 3 ijms-22-04689-t003:** Isolation, quantification and characterization of EVs in plasma of obstetric APS patients

Reference	Patient Group	Control Group	Isolation Method	Characterization Method	Type of EVs	Main Findings
Alijotas-Reig et al., 2011 [83]	9 pregnant obstetric APS	40 aPL− pregnant women with a history of pregnancy loss	Sodium citrate. 1500× *g* (15 min), 13,000× *g* (2 min).	FC: Positive for annexin V, CD41, CD45, CD144. <1 µm (calibrated latex beads). Renumeration beads (FlowCount).	total (annexin V+) platelet (CD41+) endothelial (CD144+) leukocyte (CD45+)	No difference in levels of total, platelet, leukocyte, and endothelial EVs between pregnant APS pts. and pregnant aPL− pts. with a history of pregnancy loss.
Martinez-Zamora et al., 2015 [86]	50 non-pregnant obstetric APS	50 non-pregnant aPL− patients with a history of unexplained RM. 50 HBDs.	Sodium citrate. 2000× *g* (10 min), RT 5000× *g* (10 min), 4 °C.	Functional assay (ZYMUPHEN MP-activity).	total EVs	↑ levels of total EVs in non-pregnant APS pts. vs. HBDs ↑ levels of total EVs in non-pregnant aPL− RM pts. vs. HBDs. No differences in levels of total EVs between non-pregnant APS pts. and non-pregnant aPL− RM pts.
Breen et al., 2015 [69]	11 non-pregnant obstetric APS.	18 HBDs	Sodium citrate 2x 2000× *g* (15 min, 4 °C), for procoagulant activity additional 12,000× *g* (2 min, 4 °C).	FC: Positive for CD51, CD41, CD61, or CD105. Renumeration beads (flow count fluoroshperes).	endothelial (CD51+ or CD105+) platelet (CD41+ or CD61+)	No difference in levels of endothelial and platelet EVs in non-pregnant APS pts. vs. HBDs. No difference in the EVs procoagulant activity between non-pregnant APS pts. and HBDs.
Campello et al., 2018 [84]	11 pregnant obstetric APS	15 pregnant HBDs.	Not described	FC: Positive for PS, TF, Endoglin. Detection beads: 0.5, 0.9, 3 µm (MegaMix) Renumeration beads not defined.	endothelial and platelet (markers not defined) TF+ endoglin+	↑ levels of PS+, endoglin+, and endothelial EVs in first and second trimester obstetric APS pts. vs. pregnant HBDs. ↑ levels of PS+, endoglin+, TF+, endothelial, and platelet EVs in third trimester obstetric APS pts. vs. pregnant HBDs. ↑ levels of endoglin+, TF+, and platelet EVs in high risk (triple aPL positive) pts. compared to low risk (single aPL positive) pts.
Zhou et al., 2019 [85]	25 pregnant obstetric APS	17 pregnant HBDs.	Sodium citrate. 1500× *g* (15 min) 13,000× *g* (2 min)	FACS sorting (CytoFlex). Positive for annexin V, CD41. Detection beads: 0.1 µm, 0.3 µm, 0.5 µm and 0.9 µm (MegaMix) Renumeration (Flow count fluorospheres).	platelet (annexin V+/CD41+)	No difference in levels of platelet EVs in first trimester APS pts. vs. pregnant HBDs.

aPL: antiphospholipid antibodies; APS: antiphospholipid syndrome; EVs: extracellular vesicles; FACS: florescent-activated cell sorting; FC: flow cytometry; HBDs: healthy blood donors; PS: phosphatidylserine; pts.: patients; RM; recurrent miscarriage; TF: tissue factor; ↑: elevated levels.

Levels of endothelial EVs were increased in all trimesters of obstetric APS patients compared to healthy pregnant women. Levels of platelet EVs were not increased in the first [84,85] and second trimester, but were shown to be increased in the third trimester obstetric APS patients compared to healthy pregnant women [84] (Table 3).

Levels of coagulation molecules and molecules involved in angiogenesis expressed on EVs have been investigated on EVs from obstetric APS patients. Levels of PS- and endoglin-positive EVs were increased in the obstetric APS patients in their first and second trimesters compared to healthy pregnant women, and TF was additionally expressed on EVs from obstetric APS patients in the third trimester compared to healthy pregnant women [84]. Levels of endoglin- and TF-positive EVs as well as platelet EVs were increased in pregnant obstetric APS patients at high risk (triple aPL positive) compared to low risk (single aPL positive). In contrast, the procoagulant activity of EVs isolated from plasma of non-pregnant obstetric APS patients did not differ from those obtained from healthy non-pregnant women [69] (Table 3).

The number of studies investigating the role of EVs in obstetric APS patients is limited and heterogeneous in terms of patient population, isolation protocol, and downstream analysis. In addition, relatively small sample sizes are used. All of this limits firm conclusions about the role of EVs in obstetric APS patients. However, in general it seems that the concentration of EVs changes during pregnancy and is increased in APS patients compared to healthy blood donors. EVs carry various prothrombotic molecules, but it is not clear whether these EVs are increased and more prothrombotic in the obstetric APS group.

#### 4.3.2. Obstetric APS In Vitro (Translational) Studies

##### The Release of Different EVs Populations from aPL Stimulated Placenta

aPL stimulation of placental explants increased the concentration of trophoblastic debris compared to isotype control stimulation [87,88]. Conversely, the concentration of medium/large and sEVs was not affected by stimulation with aPL compared to isotype control stimulation, but the size of sEVs was increased [89] (Table 4). Larger EVs have been suggested to be more proinflammatory; therefore, the change in EV size rather than concentration would suggest that these EVs have distinct functions on recipient cells from EVs extruded from normal placenta. Supporting this hypothesis, aPL influence EVs cargo composition conveying information of stressed placenta to the mother [89]. Furthermore, it has been reported that stimulation of the placenta with aPL altered the proteome of released macrovesicles compared to isotype control stimulation [88]. It is clear that different EV populations are released upon placental stimulation with aPL, but further research is needed to elucidate how aPL stimulation affects the concentration, size, and composition of EVs and what downstream effects the different EVs populations have on target cells.

##### EVs Derived from aPL-Exposed Placentae or from Plasma of APS Patients Activate Endothelial Cells

Increased activation of endothelial cells was observed after stimulation with EVs from aPL-stimulated human placental explants compared to isotype control stimulation [89,90,91,92] (Table 4). Endothelial cell activation is believed to be mediated by TLR-9 signaling most likely through the mitochondrial DNA found in these EVs [89]. Interestingly, endothelial cell activation was prevented by melatonin [92] or after inhibition of aPL internalization by endothelial cells [91] (Table 4). Medium/large platelet EVs isolated from pregnant APS patients with a history of recurrent miscarriage increased the levels of ICAM-1, VCAM-1, and TNFα and increased adhesion of monocytes to endothelial cells compared to stimulation with EVs derived from healthy pregnant women. In addition, increased endothelial cell apoptosis (via the p38 pathway) and increased apoptosis, and inhibition of invasion and migration of trophoblastic cells HTR-8/SVneo were observed after stimulation with platelet EVs as compared to platelet EVs from the group of healthy pregnant women [85].

Examination of the synergistic effects of stimulating endothelial cells with aPL and trophoblastic debris showed that trophoblastic debris alone, in contrast to aPL alone, stimulated endothelial cell activation. However, when aPL were added after the initial stimulation of endothelial cells with trophoblastic debris, activation remained at high levels, whereas it decreased to baseline levels when aPL were not added [90] (Table 4).

EVs extruded from placental explants appear to activate endothelial cells and create the phenotype that predisposes to PE. Current research consists of relatively heterogeneous preanalytical and postanalytical protocols, making studies difficult to compare. Different isolation methods with different centrifugation steps will yield different EV populations that may have distinct functions. Further research is needed to obtain a clearer picture by studying different EV populations.

##### EVs Derived from aPL-Exposed Placentae Reflect ER Dysfunction and Carry Different Danger Signals

Micro- and nanovesicles isolated from aPL-stimulated placental explants contained more cell death pathway proteins (e.g., mixed lineage kinase domain like pseudokinase (MLKL), misfolded proteins, mitochondrial DNA, and possessed increased ER stress (increased heat shock protein 70 (HSP70)) [89,93]. Mitochondrial DNA serves as a danger signal that can trigger the TLR signaling pathway in recipient cells, leading to endothelial cell dysfunction and consequently contributing to the increased risk of PE in women with aPL [89]. Another alarmin, high mobility group box 1 (HMGB-1), was also found to be increased in macrovesicles extruded from placental explants treated with aPL. HMGB-1 is normally expressed in the nuclei of STB, but serves as a danger signal when translocated into the cytoplasm or released into the extracellular space [94]. These findings suggest that the increased risk of pregnancy complications in obstetric APS patients may be due, at least in part, to alterations in trophoblast function upon stimulation with aPL and, more importantly, to the resulting release of EVs with altered cargo. Accumulation of HSP70 and misfolded proteins in the cytoplasm of the STB after aPL stimulation leads to TNFα secretion, propagation of ER dysfunction, and activation of the extrinsic pathway of apoptosis. One way to remove these danger signals and avoid apoptosis is to package them in the extruded EVs. While removal of these danger signals by EVs might protect the stressed STB, the cargo is potentially toxic to the maternal cells that are the ultimate targets of these EVs [93].

**Table 4 ijms-22-04689-t004:** Isolation, quantification and characterization of EVs derived from placental explants after stimulation with aPL and their effect on endothelial cells

Reference	Cell Type	Stimulation	Isolation of EVs	Characterization of EVs	Main Findings
Chen et al. 2012 [90]	1st trimester human placental explants, HMEC-1 (stimulated with aPL and trophoblastic debris), human U937 monocytes.	Murine monoclonal anti-β2GPI (ID2, IIC5), isotype IgG, trophoblastic debris from placental explants.	Trophoblastic debris: CD45+ leukocyte depletion using magnetic beads Red blood cells removed by incubation with MilliQ water, 1300× *g*.	NA	aPL did not increase ICAM-1 expression or monocyte adhesion to HMEC-1 in the absence of trophoblastic debris. ↑ surface ICAM-1 and E-selectin expression and monocyte adhesion to HMEC-1 by trophoblastic debris is prolonged following the stimulation with aPL.
Viall et al. 2013 [91]	1st trimester human placental explants (stimulated with aPL), HMEC-1 (stimulated with trophoblastic debris).	Murine monoclonal anti-β2GPI (ID2, IIC5), isotype IgG, trophoblastic debris from placental explants.	Not described.	NA	↑ surface ICAM-1 expression on HMEC-1 after stimulation with trophoblast debris extruded from ID2 and IIC5 stimulated placental explants compared to isotype controls. ↓ levels of ICAM-1 on HMEC-1 when inhibition of aPL internalization was used.
Gysler et al., 2016 [95]	1st trimester human extravillous trophoblast cell line (HTR8).	Murine monoclonal anti-β2GPI (IIC5), control IgG.	ExoQuick	Taqman MicroRNA Assay.	↑ of mIR-146a-5p, miR-146a-3p, and miR-210 in exosomes isolated from trophoblasts after treatment with aPL compared to isotype control.
Shao et al., 2016 [94]	1st trimester human placental explants (stimulated with aPL).	Murine monoclonal anti-β2GPI (IDT2), control IgG, serum of preeclamptic pts. and healthy pregnant women.	Trophoblastic debris: 300× *g* (10 min) CD45+ leukocyte depletion using magnetic beads Red blood cells removed by incubation with MilliQ water.	Immunohistochemistry and western blotting.	↑ HMGB1 in trophoblastic debris derived from placental explants treated with aPL or patient sera compared to controls.
Tong et al., 2017 [89]	1st trimester human placental explants (stimulated with aPL), HMEC-1 (stimulated with macro-, micro-, and nanovesicles), human U937 monocytes.	Murine monoclonal anti-β2GPI (ID2), control IgG, aPL derived from 5 APS pts. and controls, macro-, micro-, and nanovesicles from placental explants.	Macrovesicles: 2000× *g*, (5 min, 4 °C.), CD45+ leukocyte depletion using magnetic beads Red blood cells removed by incubation with MilliQ water. Microvesicles: 20,000× *g* (60 min, 4 °C.), Nanovesicles: 100,000× *g* (60 min, 4 °C).	NTA, PCR	Levels of nano- and microvesicles extruded from aPL stimulated placental explants were not increased compared to controls. ↑ mean and modal size of nanovesicles extruded from human serum-derived aPL stimulation of placental explants compared to control. ↑ surface ICAM-1 expression and monocyte adhesion to HMEC-1 after stimulation with macro-, micro-, and nanovesicles extruded from ID2 stimulated placental explants compared to isotype control. ↑ of mtDNA but not nucleolar DNA in micro- and nanovesicles extruded from ID2 stimulated placental explants compared to isotype control. Micro- and nanovesicles extruded from ID2 stimulated placental explants activated HMEC-1 through TLR-9 receptor signaling.
Zhao et al., 2017 [92]	1st trimester human placental explants (stimulated with aPL), HMEC-1.	Murine monoclonal anti-β2GPI (ID2), control IgG, sera of preeclamptic pts., and healthy pregnant women.	Trophoblastic debris: 300× *g* (10 min) CD45+ leukocyte depletion using magnetic beads Red blood cells removed by incubation with MilliQ water	NA	↑ ICAM-1 on HMEC-1 after stimulation with trophoblastic debris extruded from ID2 and pts. sera stimulated placental explants compared to controls. Melatonin prevented this increase.
Zhou et al., 2019 [85]	HUVEC, HTR-8/SVneo, THP-1.	Platelet microparticles from pregnant APS RM pts. and healthy pregnant women.	NA	NA	↑ HUVEC expression levels of TNFα, ICAM-1, VCAM-1 after stimulation with platelet microparticles from APS RM pts. compared to healthy pregnant group. ↑ THP-1 adherence to HUVEC and inhibition of HUVEC tube formation after stimulation with platelet microparticles from APS RM pts. compared to healthy pregnant group. ↑ HUVEC apoptosis (via p38 MAP kinase pathway) after stimulation with platelet microparticles from APS RM pts. compared to healthy pregnant group. ↑ apoptosis and inhibition of invasion and migration of HTR-8/SVneo after stimulation with platelet microparticles from APS RM pts. compared to healthy pregnant group.
Tang et al., 2020 [93]	1st trimester human healthy term and APS placental explants (stimulated with aPL).	Murine monoclonal anti-β2GPI (ID2, IIC5), isotype control IgG.	Macrovesicles: 2000× *g*, (5 min), Microvesicles: 20,000× *g* (60 min), Nanovesicles: 100,000× *g* (60 min).	Western blotting.	↑ HSP 70 (ER stress sensor) in microvesicles and nanovesicles derived from placental explants treated with aPL compared to isotype control. ↑ levels of misfolded proteins in microvesicles and nanovesicles derived from placental explants treated with aPL compared to isotype control. ↑ MLKL in microvesicles and nanovesicles derived from placental explants treated with aPL compared to isotype control.

aPL: antiphospholipid antibodies; APS: antiphospholipid syndrome; EVs: extracellular vesicles; HBDs: healthy blood donors; HMEC-1: human microvascular endothelial cells; HMGB1: high mobility group box 1 protein; HSP70: heat shock protein 70; HUVEC: human umbilical vein endothelial cells; ICAM-1: intercellular adhesion molecule 1; IgG: immunoglobulin G; MLKL: mixed lineage kinase domain like pseudokinase; mtDNA: mitochondrial DNA; NA: not analyzed; NTA: nanoparticle tracking analysis; pts.: patients; PCR: polymerase chain reaction; p38 MAP: p38 mitogen-activated protein; RM: recurrent miscarriage; TLR: toll-like receptor; TNFα: tumor necrosis factor alpha; VCAM-1: vascular cell adhesion protein 1; ↑: elevated levels.

## 5. Conclusions and Future Challenges

In recent years, the field of EVs investigating both physiological and pathological conditions has developed tremendously. EVs have emerged as important intercellular communicators between different cells, carrying a repertoire of information in the form of proteins, lipids, and nucleic acids. Endothelial dysfunction is a hallmark of vascular disorders, and EVs have been found to play an important role in hemostasis. It is therefore not surprising that EVs are also associated with thrombotic disorders, including APS, where they are increasingly recognized as a contributing factor in the pathology of both thrombotic and obstetric clinical manifestations. Elevated levels of EVs, particularly of endothelial origin, have been detected in plasma of thrombotic APS patients, whereas no consensus has been reached regarding EVs derived from other cells. It is increasingly believed that the underlying pathologies of thrombotic and obstetric APS are different, with EVs appearing to play an important role in both. In obstetric APS, research on plasma levels of EVs is limited, but there is a trend toward increased EVs levels in these patients. The lack of standardized methods for isolation and characterization is the major challenge in extravesicular research and consequently causes the heterogeneity of current reports.

In general, there is evidence that EVs activate endothelial cells, exhibit proinflammatory and procoagulant effects, interact directly with cell receptors, and/or transfer biological material. Many questions remain unanswered about the downstream mechanisms of EVs on these cells, as well as on other cells involved in the pathology of APS. Additional in vitro and animal studies are needed to further define the effects of these vesicles on endothelial cells, monocytes, and platelets.

EVs are considered important players in the pathogenesis of many systemic autoimmune diseases and could be used as potential biomarkers of the disease activity and prognosis [96]. Phenotyping of EVs not only provides information about their cell of origin, but also about the activation status of these cells. It has been shown that platelet EVs are of particular interest in rheumatoid arthritis, as their higher levels correlate with disease severity [97,98]. In APS, EVs could potentially be used as biomarkers for risk assessment of recurrent thrombotic events, which would be highly beneficial, as APS patients receive lifelong therapy. Current APS diagnostics are based on the classification criteria, but APS is often underdiagnosed due to its clinical variability and a lack of standardization of diagnostic tests [99]. Therefore, the identification of new biomarkers that could be used for diagnosis would be of great importance. Currently, there are several ongoing studies testing EVs as biomarkers in autoimmune diseases, including rheumatic disorders [96].

The ability to protect encapsulated molecules from degradation in body fluids suggests that there is a potential for EVs as biological therapeutics or drug delivery systems. EVs are being therefore increasingly investigated for their use in therapy. Mesenchymal stem cell-derived EVs have already been used to reduce inflammation in several autoimmune animal models [100]. In addition, dendritic cell-derived EVs have been investigated for the treatment of autoimmune diseases, as they can be conditioned to an immunosuppressive phenotype [101]. On the other hand, in APS, current therapy is rather cost-effective, but EVs may be beneficial as therapeutics for certain subsets of APS patients, such as catastrophic APS, where current treatment often fails and mortality rates remain high. The use of EVs as therapeutics presents certain challenges, as large quantities of cells are required and batch-to-batch variability appears to be high. Overall, the production of EVs is more challenging compared to small molecule therapies. Artificial production of EVs seems to be promising; however, current clinical trials focus predominantly on biomarker discovery rather than treatment. It seems that the use of EVs for the treatment of autoimmune diseases, although compelling, is somehow limited [96].

In the future, a better understanding of the levels and properties of EVs in the plasma of APS patients could reveal the diagnostic potential of these vesicles and perhaps define patients at higher risk for an adverse event. They could also serve as an additional marker to subtype patients. Understanding the effect of EVs on different cells would elucidate the currently unexplained pathology of APS and clarify whether they could be used not only as biomarkers of the disease, but also as therapeutic agents.

## Figures and Tables

**Figure 1 ijms-22-04689-f001:**
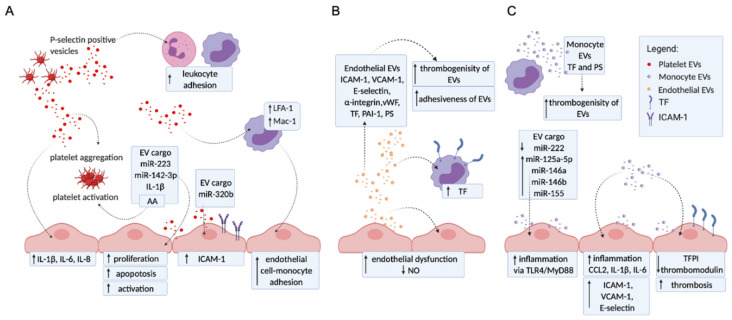
Activation of platelets, monocytes, and endothelial cells by EVs derived from different cells. Schematic representation of the potential in vitro mechanisms focusing on vascular function, inflammation, and thrombosis. (**A**) Platelet EVs (red spots) can stimulate endothelial cells and monocytes via direct interaction or by cargo delivery (miR and lipids). Furthermore, platelet EVs can also act via a feedback loop causing platelet aggregation and activation. Platelet EVs induce endothelial cell activation, proliferation, and apoptosis by the transfer or miR-223 and miR-142-3p, while ICAM-1 expression is induced by the delivery of miR-320b. Increased adhesion between endothelial cells and monocytes, as well as between leukocytes, is mediated by platelet EVs. (**B**) Endothelial EVs (orange spots) were found to have a procoagulant profile expressing vWF, TF, PAI-1, and PS as well as increased adhesive properties expressing VCAM-1, ICAM-1, E-selectin, and α-integrin. Endothelial EVs promote the procoagulant profile of monocytes by induction of TF expression on these cells and endothelial dysfunction by attenuating the production of nitric oxide from endothelial cells (**C**). Monocytes release procoagulant EVs (purple spots) that carry TF and PS. Furthermore, monocyte EVs interact with endothelial cells, causing increased expression of adhesion molecules (ICAM-1, VCAM-1, and E-selectin), increased inflammation, and procoagulant profile by reducing the expression of anticoagulant molecules (TFPI and thrombomodulin). Monocyte EVs transfer miR (miR125a-5p, miR-222, miR-146a, miR-146b, and miR-155) and induce inflammation in endothelial cells. CCL2: C-C motif chemokine ligand 2; ICAM-1: intercellular adhesion molecule 1; IL: interleukin; LFA1: lymphocyte function-associated antigen 1; Mac-1: macrophage antigen-1; mIR; micro RNA; MyD88: myeloid differentiation primary response gene 88; NO: nitric oxide; PAI-1: plasminogen activator inhibitor-1; PS: phosphatidylserine; TF: tissue factor; TLR4: toll like receptor 4; VCAM-1: vascular cell adhesion molecule 1; vWF: von Willebrand factor.

**Figure 2 ijms-22-04689-f002:**
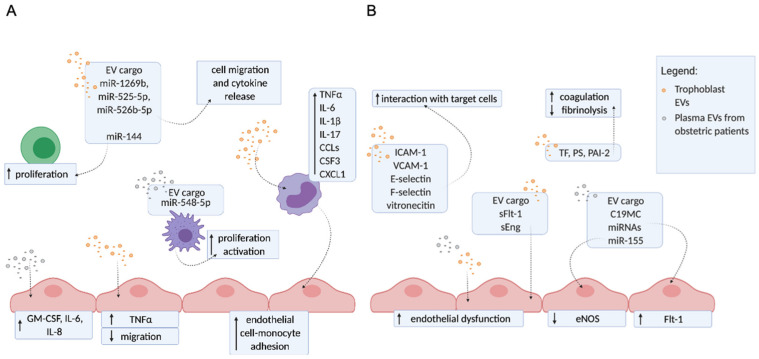
EVs isolated from plasma of patients with obstetric complications or derived from cultured trophoblasts enhance inflammation, coagulation, and endothelial dysfunction. Schematic presentation of in vitro mechanisms focusing on vascular function, inflammation, and thrombosis. (**A**) Mechanisms contributing to enhanced inflammation. EVs from trophoblasts (orange spots) carry miRNAs (miR-1269b, miR-525-5p, and miR-526b-5p) that are associated with cell migration and cytokine production in target cells and miR-144 involved in T-cell activation and proliferation. Furthermore, trophoblast EVs induce cytokine production in monocytes suppressing their chemotactic activity and motility. EVs isolated from plasma of patients with obstetric complications (grey spots) carry miR-548-5p, which was found to modulate the proliferation and activation of macrophages. EVs were also shown to interact with endothelial cells and influence the release of pro-inflammatory molecules and decrease the migration of these cells. (**B**) Mechanisms contributing to endothelial dysfunction and coagulation. EVs isolated from trophoblasts carry different adhesion molecules on their surface, potentially increasing the interactions of these EVs with target cells. In addition, on their surface, EVs carry tissue factor (TF), phosphatidylserine (PS), and plasminogen activator inhibitor-2 (PAI-2), molecules involved in increased coagulation and decreased fibrinolysis. EVs directly interact with endothelial cells causing the endothelial dysfunction, partially by the transfer of their cargo, fms-like tyrosine kinase-1 (sFlt-1) and endoglin (sEng). By the transfer of their miRNas, trophoblast EVs influence the expression levels of eNOS (miR-155) and modulate the expression of several other genes, including Flt-1, involved in endothelial dysfunction (C19MC miRNAs). CCLs: chemokine (C-C motif) ligands; CSF3: colony stimulating factor 3; CXCL1: chemokine (C-X-C motif) ligand 1; eNOS: endothelial nitric oxide synthase; GM-CSF: granulocyte-macrophage colony-stimulating factor; ILs: interleukins; ICAM-1: intercellular adhesion molecule 1; miR: micro RNA; PS: phosphatidylserine; PAI: plasminogen activator inhibitor; sFlt: soluble fms tyrosine kinase-1; sEng: soluble endoglin; TF: tissue factor; TNFα: tumor necrosis factor alpha; VCAM-1: vascular cell adhesion potein-1.

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
