# Peer review of "Extracellular Vesicles and Antiphospholipid Syndrome: State-of-the-Art and Future Challenges"

_ijms, 2021, doi:10.3390/ijms22094689_

Round 1
Reviewer 1 Report
This paperaims to describe recent reported data upon the potential role of extracellular vesicles in patients with antiphospholipid syndrome, particularly focusing on the endothelial dysfunction which favors accelerated atherosclerosis. the innovative strength of the article is based on the current description of an extremely little known role of extracellular vesicles that is actually little known and considered in the genesis of antiphospholipid syndrome. The limit is the scarce potential of application both in diagnostics and in not determining changes in the treatment envisaged for this syndrome. However, I suggest the authors to extend the review discussing and add as reference paper by Negrini et al concerning antiphospholipid syndrome that is a a comprehensive review of the antiphospholipid syndrome.
Reviewer 2 Report
This interesting review summarized recent findings on the role of extracellular vesicles (EVs) in antiphospholipid syndrome (APS), focusing on their influence on the endothelial dysfunction. The article included studies that examined EVs in the plasma of thrombotic/obstetric APS patients, and the effects of EVs isolated from antiphospholipid antibodies-stimulated endothelial cells and placental explants on other cells. The authors concluded that the field of EVs provides insight into the APS pathogenesis and serves as potential biomarkers to identify patients at risk.
The manuscript is well written in English and the content is relevant to clinical application. There is only one suggestion as follows.
In the final conclusions, the authors stated that understanding the effect of EVs on different cells would further elucidate the APS pathogenesis, and whether they could be used not only as disease biomarkers but also as therapeutic agents. Recently, there are clinical trials related to the use of EVs in autoimmune diseases therapy (Front Immunol 2021;12:566299). The authors should further extend their discussion on the therapeutic potential of EVs in APS.
Reviewer 3 Report
In this review Stok and colleagues summarize recent findings on the role of extracellular vesicles (EVs) in the antiphospholipid syndrome (APS), focusing on their role in endothelial dysfunction. They recapitulate accurately the literature on this field.
The manuscript is well organized, but some point should be addressed:
1) The paragraph on the molecular mechanisms of EVs contributing to vascular disorders should be focused on their implications on APS;
2) A previous review on EVs and APS has been already published (Clin Chem Lab Med, 55:934-943, 2017). It should be included in the reference list and considered in the text;
3) Whenever miRNAs are mentioned, they should be introduced and explained;
4) Paragraph 4.2.2: aPL are able to activate not only endothelial cells and platelets, but also monocytes, triggering a signal transduction pathway. A citation should be added.
It is important since in Figure 1 activation of all these cells by MVs is shown.
5). Some misprints should be corrected. For example: lane 115, Toll instead of tool; lane 157, important; lane 174, inflammation.
Round 2
Reviewer 3 Report
The manuscript has been improved according to the reviewers' suggestions and is now suitable for publication in IJMS.